# Mean Flows for One-step Generative Modeling

Zhengyang Geng[1]*    Mingyang Deng[2]    Xingjian Bai[2]    J. Zico Kolter[1]    Kaiming He[2]

[1]CMU    [2]MIT

## Abstract

We propose a principled and effective framework for one-step generative modeling. We introduce the notion of average velocity to characterize flow fields, in contrast to instantaneous velocity modeled by Flow Matching methods. A well-defined identity between average and instantaneous velocities is derived and used to guide neural network training. Our method, termed the *MeanFlow* model, is self-contained and requires no pre-training, distillation, or curriculum learning. MeanFlow demonstrates strong empirical performance: it achieves an FID of 3.43 with a single function evaluation (1-NFE) on ImageNet 256×256 trained from scratch, significantly outperforming previous state-of-the-art one-step diffusion/flow models. Our study substantially narrows the gap between one-step diffusion/flow models and their multi-step predecessors, and we hope it will motivate future research to revisit the foundations of these powerful models. Our code is available at https://github.com/gsunshine/meanflow.

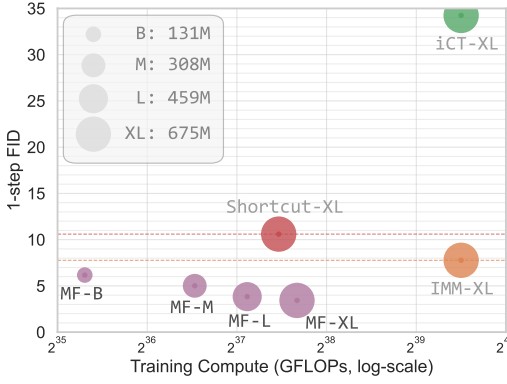
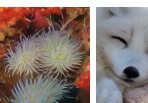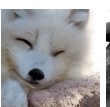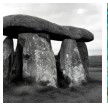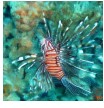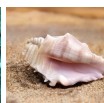

Figure 1: **One-step generation on ImageNet 256×256 from scratch**. Our *MeanFlow* (**MF**) model achieves significantly better generation quality than previous state-of-the-art one-step diffusion/flow methods. Here, iCT [43], Shortcut [12], and our MF are all **1-NFE** generation, while IMM's 1-step result [52] involves 2-NFE guidance. Detailed numbers are in Tab. 2. Images shown are generated by our 1-NFE model.

## 1 Introduction

The goal of generative modeling is to transform a prior distribution into the data distribution. Flow Matching [28, 1, 30] provides an intuitive and conceptually simple framework for constructing flow paths that transport one distribution to another. Closely related to diffusion models [42, 44, 19], Flow Matching focuses on the velocity fields that guide model training. Since its introduction, Flow Matching has seen widespread adoption in modern generative modeling [10, 33, 35].

Both Flow Matching and diffusion models perform iterative sampling during generation. Recent research has paid significant attention to few-step—and in particular, one-step, feedforward—generative models. Pioneering this direction, Consistency Models [46, 43, 14, 31] introduce a consistency constraint to network outputs for inputs sampled along the same path. Despite encouraging results, the consistency constraint is imposed as a property of the network's behavior, while the properties of the underlying ground-truth field that should guide learning remain unknown. Consequently, training can be unstable and requires a carefully designed "discretization curriculum" [46, 43, 14] to progressively constrain the time domain.

---

*Work partly done when visiting MIT.

39th Conference on Neural Information Processing Systems (NeurIPS 2025).

In this work, we propose a principled and effective framework, termed *MeanFlow*, for one-step generation. The core idea is to introduce a new ground-truth field representing the *average velocity*, in contrast to the *instantaneous velocity* typically modeled in Flow Matching. Average velocity is defined as the ratio of displacement to a time interval, with displacement given by the time integral of the instantaneous velocity. Solely from this definition, we derive a well-defined, intrinsic relation between the average and instantaneous velocities, which naturally serves as a principled basis for guiding network training.

Building on this fundamental concept, we train a neural network to directly model the average velocity field. We introduce a loss function that encourages the network to satisfy the intrinsic relation between average and instantaneous velocities. No extra consistency heuristic is needed. The existence of the ground-truth target field ensures that the optimal solution is, in principle, independent of the specific network, which in practice can lead to more robust and stable training. We further show that our framework can naturally incorporate classifier-free guidance (CFG) [18] into the target field, incurring no additional cost at sampling time when guidance is used.

Our MeanFlow models demonstrate strong empirical performance in one-step generative modeling. On ImageNet 256×256 [6], our method achieves an FID of 3.43 using 1-NFE (Number of Function Evaluations) generation. This result significantly outperforms previous state-of-the-art methods in its class by a relative margin of 50% to 70% (Fig. 1). In addition, our method stands as a self-contained generative model: it is trained entirely from scratch, without any pre-training, distillation, or curriculum learning. Our study largely closes the gap between one-step diffusion/flow models and their multi-step predecessors, and we hope it will inspire future work to reconsider the foundations of these powerful models.

## 2 Related Work

**Diffusion and Flow Matching.** Over the past decade, diffusion models [42, 44, 19, 45] have been developed into a highly successful framework for generative modeling. These models progressively add noise to clean data and train a neural network to reverse this process. This procedure involves solving stochastic differential equations (SDE), which can be reformulated as probability flow ordinary differential equations (ODE) [45, 22]. Flow Matching methods [28, 1, 30] extend this framework by modeling the velocity fields that define flow paths between distributions. Flow Matching can also be viewed as a form of continuous-time Normalizing Flows [36].

**Few-step Diffusion/Flow Models.** Reducing sampling steps has become an important consideration from both practical and theoretical perspectives. One approach is to distill a pre-trained many-step diffusion model into a few-step model, *e.g.,* [39, 13, 41] or score distillation [32, 50, 53]. Early explorations into training few-step models [46] are built upon the evolution of distillation-based methods. Meanwhile, Consistency Models (CM) [46] are developed as a standalone generative model that does not require distillation. These models impose consistency constraints on network outputs at different time steps, encouraging them to produce the same endpoints along the trajectory. Various consistency models and training strategies [46, 43, 14, 31, 49, 16] have been investigated.

Several recent works have focused on characterizing diffusion-/flow-based quantities with respect to *two* time-dependent variables. Consistency Trajectory Models [23] extend to two time variables, using a training-time ODE solver for target computation, which is computationally expensive. In [2], a Flow Map is defined as the flow's integral between two time steps. Compared to the average velocity we are based on, [23, 2] are analogous to displacement. Shortcut Models [12] introduce a self-consistency loss function in addition to Flow Matching, which captures relationships between the flows at different discrete time intervals. Inductive Moment Matching [52] models the self-consistency of stochastic interpolants at different time steps.

## 3 Background: Flow Matching

Flow Matching [28, 30, 1] is a family of generative models that learn to match the flows, represented by velocity fields, between two probabilistic distributions. Formally, given data $x \sim p_{\text{data}}(x)$ and prior $\epsilon \sim p_{\text{prior}}(\epsilon)$, a flow path can be constructed as $z_t = a_t x + b_t \epsilon$ with time $t$, where $a_t$ and $b_t$ are predefined schedules. The velocity $v_t$ is defined as $v_t = z_t' = a_t' x + b_t' \epsilon$, where $'$ denotes the time derivative. This velocity is referred to as the *conditional* velocity in [28], denoted by $v_t = v_t(z_t \mid x)$. See Fig. 2 left. A commonly used schedule is $a_t = 1 - t$ and $b_t = t$, which leads to $v_t = \epsilon - x$.

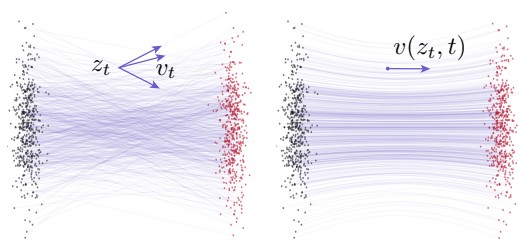

Figure 2: **Velocity fields in Flow Matching** [28]. **Left**: *conditional* flows [28]. A given $z_t$ can arise from different $(x, \epsilon)$ pairs, resulting in different conditional velocities $v_t$. **Right**: *marginal* flows [28], obtained by marginalizing over all possible conditional velocities. The marginal velocity field serves as the underlying ground-truth field for network training. All velocities shown here are essentially ***instantaneous*** velocities. Illustration follows [11]. (*Gray dots: samples from prior; red dots: samples from data.*)

Because a given $z_t$ and its $v_t$ can arise from different $x$ and $\epsilon$, Flow Matching essentially models the expectation over all possibilities, called the *marginal* velocity [28] (Fig. 2 right):

$$v(z_t, t) \triangleq \mathbb{E}_{p_t(v_t|z_t)}[v_t]. \tag{1}$$

A neural network $v_\theta$ parameterized by $\theta$ is learned to fit the marginal velocity field: $\mathcal{L}_{\text{FM}}(\theta) = \mathbb{E}_{t,p_t(z_t)}\|v_\theta(z_t, t) - v(z_t, t)\|^2$. Although computing this loss function is infeasible due to the marginalization in Eq. (1), it is proposed to instead evaluate the *conditional* Flow Matching loss [28]: $\mathcal{L}_{\text{CFM}}(\theta) = \mathbb{E}_{t,x,\epsilon}\|v_\theta(z_t, t) - v_t(z_t \mid x)\|^2$, where the target $v_t$ is the conditional velocity. Minimizing $\mathcal{L}_{\text{CFM}}$ is equivalent to minimizing $\mathcal{L}_{\text{FM}}$ [28].

Given a marginal velocity field $v(z_t, t)$, samples are generated by solving an ODE for $z_t$:

$$\frac{d}{dt}z_t = v(z_t, t) \tag{2}$$

starting from $z_1 = \epsilon \sim p_{\text{prior}}$. The solution can be written as: $z_r = z_t - \int_r^t v(z_\tau, \tau)d\tau$, where we use $r$ to denote another time step. In practice, this integral is approximated numerically over discrete time steps. For example, the Euler method, a first-order ODE solver, computes each step as: $z_{t_{i+1}} = z_{t_i} + (t_{i+1} - t_i)v(z_{t_i}, t_i)$. Higher-order solvers can also be applied.

It is worth noting that even when the conditional flows are designed to be straight ("rectified") [28, 30], the marginal velocity field (Eq. (1)) typically induces a curved trajectory. See Fig. 2 for illustration. We also emphasize that this non-straightness is not only a result of neural network approximation, but rather arises from the underlying ground-truth marginal velocity field. When applying coarse discretizations over curved trajectories, numerical ODE solvers lead to inaccurate results.

## 4 MeanFlow Models

### 4.1 Mean Flows

The core idea of our approach is to introduce a new field representing *average velocity*, whereas the velocity modeled in Flow Matching represents the *instantaneous velocity*.

**Average Velocity.** We define average velocity as the displacement between two time steps $t$ and $r$ (obtained by integration) divided by the time interval. Formally, the average velocity $u$ is:

$$u(z_t, r, t) \triangleq \frac{1}{t-r} \int_r^t v(z_\tau, \tau)d\tau. \tag{3}$$

To emphasize the conceptual difference, throughout this paper, we use the notation $u$ to denote average velocity, and $v$ to denote instantaneous velocity. $u(z_t, r, t)$ is a field that is jointly dependent on $(r, t)$. The field of $u$ is illustrated in Fig. 3. Note that in general, the average velocity $u$ is the result of a *functional* of the instantaneous velocity $v$: that is, $u = \mathcal{F}[v] \triangleq \frac{1}{t-r} \int_r^t v d\tau$. It is a field induced by $v$, not depending on any neural network. Conceptually, just as the instantaneous velocity $v$ serves as the ground-truth field in Flow Matching, the average velocity $u$ in our formulation provides an underlying ground-truth field for learning.

By definition, the field of $u$ satisfies certain boundary conditions and "consistency" constraints (generalizing the terminology of [46]). As $r \to t$, we have: $\lim_{r \to t} u = v$. Moreover, a form of "consistency" is naturally satisfied: taking one larger step over $[r, t]$ is "consistent" with taking two smaller consecutive steps over $[r, s]$ and $[s, t]$, for any intermediate time $s$. To see this, observe that $(t-r)u(z_t, r, t) = (s-r)u(z_s, r, s) + (t-s)u(z_t, s, t)$, which follows directly from the additivity

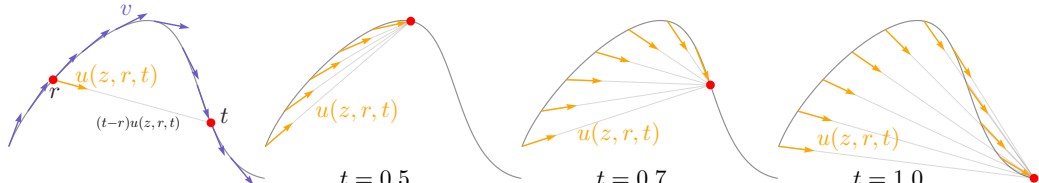

Figure 3: **The field of *average velocity* $u(z, r, t)$. Leftmost**: While the *instantaneous velocity* $v$ determines the tangent direction of the path, the average velocity $u(z, r, t)$, defined in Eq. (3), is generally not aligned with $v$. The average velocity is aligned with the *displacement*, which is $(t - r)u(z, r, t)$. **Right three subplots**: The field $u(z, r, t)$ is conditioned on both $r$ and $t$, and is shown here for $t = 0.5$, $0.7$, and $1.0$.

of the integral: $\int_r^t v d\tau = \int_r^s v d\tau + \int_s^t v d\tau$. Thus, a network that accurately approximates the true $u$ is expected to satisfy the consistency relation inherently, *without* the need for explicit constraints.

The ultimate aim of our MeanFlow model will be to approximate the average velocity using a neural network $u_\theta(z_t, r, t)$. This has the notable advantage that, assuming we approximate this quantity accurately, we can approximate the entire flow path using a *single* evaluation of $u_\theta(\epsilon, 0, 1)$. In other words, and as we will also demonstrate empirically, the approach is much more amenable to single or few-step generation, as it does not need to explicitly approximate a time integral at inference time, which was required when modeling instantaneous velocity. However, directly using the average velocity defined by Eq. (3) as ground truth for training a network is intractable, as it requires evaluating an integral during training. Our key insight is that the definitional equation of average velocity can be manipulated to construct an optimization target that is ultimately amenable to training, even when only the *instantaneous* velocity is accessible.

**The MeanFlow Identity.** To have a formulation amenable to training, we rewrite Eq. (3) as:

$$(t - r)u(z_t, r, t) = \int_r^t v(z_\tau, \tau)d\tau. \tag{4}$$

Now we differentiate both sides with respect to $t$, treating $r$ as independent of $t$. This leads to:

$$\frac{d}{dt}(t - r)u(z_t, r, t) = \frac{d}{dt}\int_r^t v(z_\tau, \tau)d\tau \implies u(z_t, r, t) + (t - r)\frac{d}{dt}u(z_t, r, t) = v(z_t, t), \tag{5}$$

where the manipulation of the left hand side employs the product rule and the right hand side uses the fundamental theorem of calculus[2]. Rearranging terms, we obtain the identity:

$$\boxed{\underbrace{u(z_t, r, t)}_{\text{average vel.}} = \underbrace{v(z_t, t)}_{\text{instant. vel.}} - (t - r)\underbrace{\frac{d}{dt}u(z_t, r, t)}_{\text{time derivative}}} \tag{6}$$

We refer to this equation as the "**MeanFlow Identity**", which describes the relation between $v$ and $u$. It is easy to show that Eq. (6) and Eq. (4) are equivalent (see Appendix B.3).

The right hand side of Eq. (6) provides a "target" form for $u(z_t, r, t)$, which we will leverage to construct a loss function to train a neural network. To serve as a suitable target, we must also further decompose the time derivative term, which we discuss next.

**Computing Time Derivative.** To compute the $\frac{d}{dt}u$ term in Eq. (6), note that $\frac{d}{dt}$ denotes a total derivative, which can be expanded in terms of partial derivatives:

$$\frac{d}{dt}u(z_t, r, t) = \frac{dz_t}{dt}\partial_z u + \frac{dr}{dt}\partial_r u + \frac{dt}{dt}\partial_t u. \tag{7}$$

With $\frac{dz_t}{dt} = v(z_t, t)$ (see Eq. (2)), $\frac{dr}{dt} = 0$, and $\frac{dt}{dt} = 1$, we have another relation between $u$ and $v$:

$$\boxed{\frac{d}{dt}u(z_t, r, t) = v(z_t, t)\partial_z u + \partial_t u,} \tag{8}$$

---

[2]If $r$ depends on $t$, the Leibniz rule [26] gives: $\frac{d}{dt}\int_r^t v(z_\tau, \tau)d\tau = v(z_t, t) - v(z_r, r)\frac{dr}{dt}$.

This equation shows that the total derivative is given by the Jacobian-vector product (JVP) between $[\partial_z u, \partial_r u, \partial_t u]$ (the Jacobian matrix of the function $u$) and the tangent vector $[v, 0, 1]$. In modern libraries, this can be efficiently computed by the `jvp` interface, such as `torch.func.jvp` in PyTorch or `jax.jvp` in JAX, which we discuss later.

**Training with Average Velocity.** Up to this point, the formulations are independent of any network parameterization. We now introduce a model to learn $u$. Formally, we parameterize a network $u_\theta$ and encourage it to satisfy the MeanFlow Identity (Eq. (6)). Specifically, we minimize this objective:

$$\mathcal{L}(\theta) = \mathbb{E}\big\|u_\theta(z_t, r, t) - \mathrm{sg}(u_{\text{tgt}})\big\|_2^2, \tag{9}$$

$$\text{where} \qquad u_{\text{tgt}} = v(z_t, t) - (t - r)\left(v(z_t, t)\partial_z u_\theta + \partial_t u_\theta\right), \tag{10}$$

The term $u_{\text{tgt}}$ serves as the *effective regression target*, which is driven by Eq. (6). This target uses the instantaneous velocity $v$ as the only ground-truth signal; no integral computation is needed. While the target should involve derivatives of $u$ (that is, $\partial u$), they are replaced by their parameterized counterparts (that is, $\partial u_\theta$). In the loss function, a stop-gradient (sg) operation is applied on the target $u_{\text{tgt}}$, following common practice [46, 43, 14, 31, 12]: in our case, it eliminates the need for "double backpropagation" through the Jacobian-vector product, thereby avoiding higher-order optimization. Despite these practices for *optimizability*, if $u_\theta$ were to achieve zero loss, it is easy to show that it would satisfy the MeanFlow Identity (Eq. (6)), and thus satisfy the original definition (Eq. (3)).

The velocity $v(z_t, t)$ in Eq. (10) is the marginal velocity in Flow Matching [28] (see Fig. 2 right). We follow [28] to replace it with the conditional velocity (Fig. 2 left). With this, the target is:

$$u_{\text{tgt}} = v_t - (t - r)\big(v_t\partial_z u_\theta + \partial_t u_\theta\big). \tag{11}$$

Recall that $v_t = a_t' x + b_t' \epsilon$ is the conditional velocity [28], and by default, $v_t = \epsilon - x$.

Pseudocode for minimizing the loss function Eq. (9) is presented in Alg. 1. Overall, our method is conceptually simple: it behaves similarly to Flow Matching, with the key difference that the matching target is modified by $-(t-r)\left(v_t\partial_z u_\theta + \partial_t u_\theta\right)$, arising from our consideration of the average velocity. In particular, note that if we were to restrict to the condition $t = r$, then the second term vanishes, and the method would exactly match standard Flow Matching.

In Alg. 1, the `jvp` operation is highly efficient. In essence, computing $\frac{d}{dt}u$ via `jvp` requires only a single backward pass, similar to standard backpropagation in neural networks. Because $\frac{d}{dt}u$ is part of the target $u_{\text{tgt}}$ and thus subject to stopgrad (*w.r.t.* $\theta$), the backpropagation for neural network optimization (*w.r.t.* $\theta$) treats $\frac{d}{dt}u$ as a constant, *incurring no higher-order gradient computation.* Consequently, `jvp` introduces only a single extra backward pass, and its cost is comparable to that of backpropagation. In our JAX implementation of Alg. 1, the overhead is less than 20% of the total training time (see appendix).

**Sampling.** Sampling using a MeanFlow model is performed simply by replacing the time integral with the average velocity:

$$z_r = z_t - (t - r)u(z_t, r, t) \tag{12}$$

In the case of 1-step sampling, we simply have $z_0 = z_1 - u(z_1, 0, 1)$, where $z_1 = \epsilon \sim p_{\text{prior}}(\epsilon)$. Alg. 2 provides the pseudocode. Although one-step sampling is the main focus on this work, we emphasize that few step sampling is also straightforward given this equation.

---

**Algorithm 1** MeanFlow: Training.

Note: in PyTorch and JAX, `jvp` returns the function output and JVP.

```
# fn(z, r, t): function to predict u
# x: training batch

t, r = sample_t_r()
e = randn_like(x)

z = (1 - t) * x + t * e
v = e - x

u, dudt = jvp(fn, (z, r, t), (v, 0, 1))

u_tgt = v - (t - r) * dudt
error = u - stopgrad(u_tgt)

loss = metric(error)
```

---

**Algorithm 2** MeanFlow: 1-step Sampling

```
e = randn(x_shape)
x = e - fn(e, r=0, t=1)
```

---

**Relation to Prior Work.** While related to previous one-step generative models [46, 43, 14, 31, 49, 23, 12, 52], our method provides a more principled framework. At the core of our method is the

functional relationship between two underlying fields $v$ and $u$, which naturally leads to the MeanFlow Identity that $u$ must satisfy (Eq. (6)). This identity does not depend on the introduction of neural networks. In contrast, prior works typically rely on extra consistency constraints, imposed on the behavior of the neural network. Consistency Models [46, 43, 14, 31] are focused on paths anchored at the data side: in our notations, this corresponds to fixing $r \equiv 0$ for any $t$. As a result, Consistency Models are conditioned on a single time variable, unlike ours. On the other hand, the Shortcut [12] and IMM [52] models are conditioned on two time variables: they introduce additional two-time self-consistency constraints. In contrast, our method is solely driven by the definition of average velocity, and the MeanFlow Identity (Eq. (6)) used for training is naturally derived from this definition, with no extra assumption.

## 4.2 Mean Flows with Guidance

Our method naturally supports classifier-free guidance (CFG) [18]. Rather than naïvely applying CFG at sampling time, which would double NFE, we treat CFG as a property of the underlying ground-truth fields. This formulation allows us to enjoy the benefits of CFG while maintaining the 1-NFE behavior during sampling.

**Ground-truth Fields.** We construct a new ground-truth field $v^{\mathrm{cfg}}$:

$$v^{\mathrm{cfg}}(z_t, t \mid \mathbf{c}) \triangleq \omega\, v(z_t, t \mid \mathbf{c}) + (1 - \omega)\, v(z_t, t), \tag{13}$$

which is a linear combination of a class-conditional and a class-unconditional field:

$$v(z_t, t \mid \mathbf{c}) \triangleq \mathbb{E}_{p_t(v_t \mid z_t, \mathbf{c})}[v_t] \quad \text{and} \quad v(z_t, t) \triangleq \mathbb{E}_{\mathbf{c}}[v(z_t, t \mid \mathbf{c})], \tag{14}$$

where $v_t$ is the conditional velocity [28] (more precisely, *sample*-conditional velocity in this context).

Following the spirit of MeanFlow, we introduce the average velocity $u^{\mathrm{cfg}}$ corresponding to $v^{\mathrm{cfg}}$. As per the MeanFlow Identity (Eq. (6)), $u^{\mathrm{cfg}}$ satisfies:

$$u^{\mathrm{cfg}}(z_t, r, t \mid \mathbf{c}) = v^{\mathrm{cfg}}(z_t, t \mid \mathbf{c}) - (t - r)\frac{d}{dt}u^{\mathrm{cfg}}(z_t, r, t \mid \mathbf{c}). \tag{15}$$

Again, $v^{\mathrm{cfg}}$ and $u^{\mathrm{cfg}}$ are underlying ground-truth fields that do not depend on neural networks. Here, $v^{\mathrm{cfg}}$, as defined in Eq. (13), can be rewritten as:

$$v^{\mathrm{cfg}}(z_t, t \mid \mathbf{c}) = \omega\, v(z_t, t \mid \mathbf{c}) + (1 - \omega)\, u^{\mathrm{cfg}}(z_t, t, t), \tag{16}$$

where we leverage the relation[3]: $v(z_t, t) = v^{\mathrm{cfg}}(z_t, t)$, as well as $v^{\mathrm{cfg}}(z_t, t) = u^{\mathrm{cfg}}(z_t, t, t)$.

**Training with Guidance.** With Eq. (15) and Eq. (16), we construct a network and its learning target. We directly parameterize $u^{\mathrm{cfg}}$ by a function $u^{\mathrm{cfg}}_\theta$. Based on Eq. (15), we obtain the objective:

$$\mathcal{L}(\theta) = \mathbb{E}\big\| u^{\mathrm{cfg}}_\theta(z_t, r, t \mid \mathbf{c}) - \mathrm{sg}(u_{\mathrm{tgt}}) \big\|_2^2, \tag{17}$$

$$\text{where} \quad u_{\mathrm{tgt}} = \tilde{v}_t - (t - r)\big(\tilde{v}_t \partial_z u^{\mathrm{cfg}}_\theta + \partial_t u^{\mathrm{cfg}}_\theta\big). \tag{18}$$

This formulation is similar to Eq. (9), with the only difference that it has a modified $\tilde{v}_t$:

$$\tilde{v}_t \triangleq \omega\, v_t + (1 - \omega)\, u^{\mathrm{cfg}}_\theta(z_t, t, t), \tag{19}$$

which is driven by Eq. (16): the term $v(z_t, t \mid \mathbf{c})$ in Eq. (16), which is the marginal velocity, is replaced by the (sample-)conditional velocity $v_t$, following [28]. If $\omega = 1$, this loss function degenerates to the no-CFG case in Eq. (9).

To expose the network $u^{\mathrm{cfg}}_\theta$ in Eq. (17) to class-unconditional inputs, we drop the class condition with 10% probability, following [18]. Driven by a similar motivation, we can also expose $u^{\mathrm{cfg}}_\theta(z_t, t, t)$ in Eq. (19) to both class-unconditional and class-conditional versions: the details are in Appendix B.1.

**Single-NFE Sampling with CFG.** In our formulation, $u^{\mathrm{cfg}}_\theta$ directly models $u^{\mathrm{cfg}}$, which is the average velocity induced by the CFG velocity $v^{\mathrm{cfg}}$ (Eq. (13)). As a result, no linear combination is required during sampling: we directly use $u^{\mathrm{cfg}}_\theta$ for one-step sampling (see Alg. 2), with only a single NFE. This formulation preserves the desirable single-NFE behavior.

---

[3]Observe that: $v^{\mathrm{cfg}}(z_t, t) \triangleq \mathbb{E}_{\mathbf{c}}[v^{\mathrm{cfg}}(z_t, t \mid \mathbf{c})] = \omega\, \mathbb{E}_{\mathbf{c}}[v(z_t, t \mid \mathbf{c})] + (1 - \omega)\, v(z_t, t) = v(z_t, t)$.

### 4.3 Design Decisions

**Loss Metrics.** In Eq. (9), the metric considered is the squared L2 loss. Following [46, 43, 14], we investigate different loss metrics. In general, we consider the loss function in the form of $\mathcal{L} = \|\Delta\|_2^{2\gamma}$, where $\Delta$ denotes the regression error. It can be proven (see [14]) that minimizing $\|\Delta\|_2^{2\gamma}$ is equivalent to minimizing the squared L2 loss $\|\Delta\|_2^2$ with "*adapted loss weights*". Details are in the appendix. In practice, we set the weight as $w = 1/(\|\Delta\|_2^2 + c)^p$, where $p = 1 - \gamma$ and $c > 0$ (*e.g.,* $10^{-3}$). The adaptively weighted loss is $\mathrm{sg}(w) \cdot \mathcal{L}$, with $\mathcal{L} = \|\Delta\|_2^2$. If $p = 0.5$, this is similar to the Pseudo-Huber loss in [43]. We compare different $p$ values in experiments.

**Sampling Time Steps** $(r, t)$. We sample the two time steps $(r, t)$ from a predefined distribution. We investigate two types of distributions: (i) a uniform distribution, $\mathcal{U}(0, 1)$, and (ii) a logit-normal (lognorm) distribution [10], where a sample is first drawn from a normal distribution $\mathcal{N}(\mu, \sigma)$ and then mapped to $(0, 1)$ using the logistic function. Given a sampled pair, we assign the larger value to $t$ and the smaller to $r$. We set a certain portion of random samples with $r = t$.

**Conditioning on** $(r, t)$. We use positional embedding [48] to encode the time variables, which are then combined and provided as the conditioning of the neural network. We note that although the field is parameterized by $u_\theta(z_t, r, t)$, it is not necessary for the network to directly condition on $(r, t)$. For example, we can let the network directly condition on $(t, \Delta t)$, with $\Delta t = t - r$. In this case, we have $u_\theta(\cdot, r, t) \triangleq \mathtt{net}(\cdot, t, t - r)$ where $\mathtt{net}$ is the network. The JVP computation is always *w.r.t.* the function $u_\theta(\cdot, r, t)$. We compare different forms of conditioning in experiments.

## 5 Experiments

**Experiment Setting.** We conduct our major experiments on ImageNet [6] generation at $256{\times}256$ resolution. We evaluate Fréchet Inception Distance (FID) [17] on 50K generated images. We examine the number of function evaluations (NFE) and study 1-NFE generation by default. Following [34, 12, 52], we implement our models on the latent space of a pre-trained VAE tokenizer [37]. For $256{\times}256$ images, the tokenizer produces a latent space of $32{\times}32{\times}4$, which is the input to the model. Our models are all trained *from scratch*. Implementation details are in Appendix A.

In our ablation study, we use the ViT-B/4 architecture (namely, "Base" size with a patch size of 4) [8] as developed in [34], trained for 80 epochs (400K iterations). As a reference, DiT-B/4 in [34] has 68.4 FID, and SiT-B/4 [33] (in our reproduction) has 58.9 FID, both using 250-NFE sampling.

### 5.1 Ablation Study

We investigate the model properties in Tab. 1, analyzed next:

**From Flow Matching to Mean Flows.** Our method can be viewed as Flow Matching with a modified target (Alg. 1), and it reduces to standard Flow Matching when $r$ always equals $t$. Tab. 1a compares the ratio of sampling $r \neq t$. A 0% ratio of $r \neq t$ (reducing to Flow Matching) fails to produce reasonable results for 1-NFE generation. A non-zero ratio of $r \neq t$ enables MeanFlow to take effect, yielding meaningful results under 1-NFE generation. We observe that the model balances learning the instantaneous velocity ($r = t$) *vs.* propagating into $r \neq t$ via the modified target. Here, the optimal FID is achieved at a ratio of 25%, and a ratio of 100% also yields a valid result.

**JVP Computation.** The JVP operation Eq. (8) serves as the core relation that connects all $(r, t)$ coordinates. In Tab. 1b, we conduct a *destructive* comparison in which incorrect JVP computation is intentionally performed. It shows that meaningful results are achieved only when the JVP computation is correct. Notably, the JVP tangent along $\partial_z u$ is $d$-dimensional, where $d$ is the data dimension (here, $32{\times}32{\times}4$), and the tangents along $\partial_r u$ and $\partial_t u$ are *one-dimensional*. Nevertheless, these two time variables determine the field $u$, and their roles are therefore critical even though they are only one-dimensional.

**Conditioning on** $(r, t)$. As discussed in Sec. 4.3, we can represent $u_\theta(z, r, t)$ by various forms of explicit positional embedding, *e.g.,* $u_\theta(\cdot, r, t) \triangleq \mathtt{net}(\cdot, t, t - r)$. Tab. 1c compares these variants. Tab. 1c shows that *all variants* of $(r, t)$ embeddings studied yield meaningful 1-NFE results, demon-

| % of $r \neq t$ | FID, 1-NFE |
|---|---|
| 0% (= FM) | 328.91 |
| 25% | **61.06** |
| 50% | 63.14 |
| 100% | 67.32 |

(a) **Ratio of sampling** $r \neq t$. The 0% entry reduces to the standard Flow Matching baseline.

| jvp tangent | FID, 1-NFE |
|---|---|
| $(v, 0, 1)$ | **61.06** |
| $(v, 0, 0)$ | 268.06 |
| $(v, 1, 0)$ | 329.22 |
| $(v, 1, 1)$ | 137.96 |

(b) **JVP computation**. The correct jvp tangent is $(v, 0, 1)$ for Jacobian $(\partial_z u, \partial_r u, \partial_t u)$.

| pos. embed | FID, 1-NFE |
|---|---|
| $(t, r)$ | 61.75 |
| $(t, t-r)$ | **61.06** |
| $(t, r, t-r)$ | 63.98 |
| $t-r$ only | 63.13 |

(c) **Positional embedding**. The network is conditioned on the embeddings applied to the specified variables.

| $t, r$ sampler | FID, 1-NFE |
|---|---|
| uniform(0, 1) | 65.90 |
| lognorm(–0.2, 1.0) | 63.83 |
| lognorm(–0.2, 1.2) | 64.72 |
| lognorm(–0.4, 1.0) | **61.06** |
| lognorm(–0.4, 1.2) | 61.79 |

(d) **Time samplers**. $t$ and $r$ are sampled from the specific sampler.

| $p$ | FID, 1-NFE |
|---|---|
| 0.0 | 79.75 |
| 0.5 | 63.98 |
| 1.0 | **61.06** |
| 1.5 | 66.57 |
| 2.0 | 69.19 |

(e) **Loss metrics**. $p=0$ is squared L2 loss. $p=0.5$ is Pseudo-Huber loss.

| $\omega$ | FID, 1-NFE |
|---|---|
| 1.0 (w/o cfg) | 61.06 |
| 1.5 | 33.33 |
| 2.0 | 20.15 |
| 3.0 | **15.53** |
| 5.0 | 20.75 |

(f) **CFG scale**. Our method supports 1-NFE CFG sampling.

Table 1: **Ablation study on 1-NFE ImageNet 256×256 generation.** FID-50K is evaluated. Default configurations are marked in  gray : B/4 backbone, 80-epoch training from scratch.

strating the effectiveness of MeanFlow as a framework. Embedding $(t, t - r)$, that is, time and interval, achieves the best result, while directly embedding $(r, t)$ performs almost as well. Notably, even embedding only the interval $t - r$ yields reasonable results.

**Time Samplers.** Prior work [10] has shown that the distribution used to sample $t$ influences the generation quality. We study the distribution used to sample $(r, t)$ in Tab. 1d. Note that $(r, t)$ are first sampled independently, followed by a post-processing step that enforces $t > r$ by swapping and then caps the proportion of $r \neq t$ to a specified ratio. Tab. 1d reports that a logit-normal sampler performs the best, consistent with observations on Flow Matching [10].

**Loss Metrics.** It has been reported [43] that the choice of loss metrics strongly impacts the performance of few-/one-step generation. We study this aspect in Tab. 1e. Our loss metric is implemented via adaptive loss weighting [14] with power $p$ (Sec. 4.3). Tab. 1e shows that $p = 1$ achieves the best result, whereas $p = 0.5$ (similar to Pseudo-Huber loss [43]) also performs competitively. The standard squared L2 loss (here, $p = 0$) underperforms compared to other settings, but still produces meaningful results, consistent with observations in [43].

**Guidance Scale.** Tab. 1f reports the results with CFG. Consistent with observations in multi-step generation [34], CFG substantially improves generation quality in our 1-NFE setting too. We emphasize that our CFG formulation (Sec. 4.2) naturally supports 1-NFE sampling.

**Scalability.** Fig. 4 presents the 1-NFE FID results of MeanFlow across larger model sizes and different training durations. Consistent with the behavior of Transformer-based diffusion/flow models (DiT [34] and SiT [33]), MeanFlow models exhibit promising scalability for 1-NFE generation.

## 5.2 Comparisons with Prior Work

**ImageNet 256×256 Comparisons.** In Fig. 1 we compare with previous one-step diffusion/flow models, which are also summarized in Tab. 2 (left). Overall, MeanFlow largely outperforms previous methods in its class: it achieves **3.43** FID, which is an over 50% relative improvement *vs.* IMM's one-step result of 7.77 [52]; if we compare only 1-NFE (not just one-step) generation, MeanFlow has nearly 70% relative improvement *vs.* the previous state-of-the-art (10.60, Shortcut [12]). Our method largely closes the gap between one-step and many-step diffusion/flow models.

In 2-NFE generation, our method achieves an FID of **2.20** (Tab. 2, bottom left). This result is on par with the leading baselines of many-step diffusion/flow models, namely, DiT [34] (FID 2.27) and SiT [33] (FID 2.15), both having an NFE of 250×2 (Tab. 2, right), under the same XL/2 backbone. Our

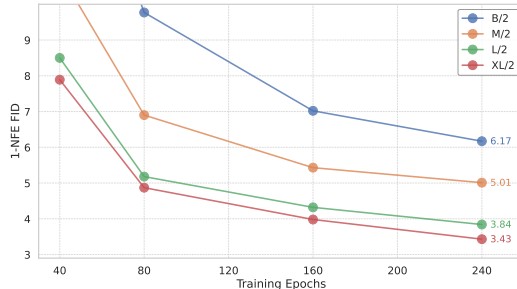

Figure 4: **Scalability of MeanFlow models on ImageNet 256×256.** 1-NFE generation FID is reported. All models are trained from scratch. CFG is applied while maintaining the 1-NFE sampling behavior. Our method exhibits promising scalability with respect to model size.

| method | params | NFE | FID |
|---|---|---|---|
| *1-NFE diffusion/flow from scratch* | | | |
| iCT-XL/2 [43][†] | 675M | 1 | 34.24 |
| Shortcut-XL/2 [12] | 675M | 1 | 10.60 |
| MeanFlow-B/2 | 131M | 1 | 6.17 |
| MeanFlow-M/2 | 308M | 1 | 5.01 |
| MeanFlow-L/2 | 459M | 1 | 3.84 |
| MeanFlow-XL/2 | 676M | 1 | **3.43** |
| *2-NFE diffusion/flow from scratch* | | | |
| iCT-XL/2 [43][†] | 675M | 2 | 20.30 |
| IMM-XL/2 [52] | 675M | 1×2 | 7.77 |
| MeanFlow-XL/2 | 676M | 2 | 2.93 |
| MeanFlow-XL/2+ | 676M | 2 | **2.20** |

| method | params | NFE | FID |
|---|---|---|---|
| *GANs* | | | |
| BigGAN [4] | 112M | 1 | 6.95 |
| GigaGAN [21] | 569M | 1 | 3.45 |
| StyleGAN-XL [40] | 166M | 1 | 2.30 |
| **autoregressive/masking** | | | |
| AR w/ VQGAN [9] | 227M | 1024 | 26.52 |
| MaskGIT [5] | 227M | 8 | 6.18 |
| VAR-$d$30 [47] | 2B | 10×2 | 1.92 |
| MAR-H [27] | 943M | 256×2 | 1.55 |
| *diffusion/flow* | | | |
| ADM [7] | 554M | 250×2 | 10.94 |
| LDM-4-G [37] | 400M | 250×2 | 3.60 |
| SimDiff [20] | 2B | 512×2 | 2.77 |
| DiT-XL/2 [34] | 675M | 250×2 | 2.27 |
| SiT-XL/2 [33] | 675M | 250×2 | 2.06 |
| SiT-XL/2+REPA [51] | 675M | 250×2 | **1.42** |

Table 2: **Class-conditional generation on ImageNet-256×256**. All entries are reported with CFG, when applicable. **Left**: **1-NFE** and **2-NFE** diffusion/flow models trained from scratch. **Right**: Other families of generative models as a reference. In both tables, "×2" indicates that CFG incurs an NFE of 2 per sampling step. Our MeanFlow models are all trained for 240 epochs, except that "MeanFlow-XL/2+" is trained for more epochs and with configurations selected for longer training, specified in appendix. [†]: iCT [43] results are reported by [52].

results suggest that *few-step diffusion/flow models can rival their many-step predecessors*. Orthogonal improvements, such as REPA [51], are applicable, which are left for future work.

Notably, our method is self-contained and trained entirely ***from scratch***. It achieves the strong results *without* using any pre-training, distillation, or the curriculum learning adopted in [43, 14, 31].

**CIFAR-10 Comparisons.** We report unconditional generation results on CIFAR-10 [25] (32×32) in Tab. 3. FID-50K is reported with 1-NFE sampling. All entries are with the same U-net [38] developed from [44] (∼55M), applied directly on the pixel space. All other competitors are with the EDM-style pre-conditioner [22], and ours has no preconditioner. Implementation details are in the appendix. On this dataset, our method is competitive with prior approaches.

| method | precond | NFE | FID |
|---|---|---|---|
| iCT [43] | EDM | 1 | **2.83** |
| ECT [14] | EDM | 1 | 3.60 |
| sCT [31] | EDM | 1 | 2.97 |
| IMM [52] | EDM | 1 | 3.20 |
| MeanFlow | none | 1 | 2.92 |

Table 3: **Unconditional CIFAR-10.**

# 6 Conclusion

We have presented MeanFlow, a principled and effective framework for one-step generation. Broadly speaking, the scenario considered in this work is related to *multi-scale* simulation problems in physics that may involve a range of scales, lengths, and resolutions, in space and time. Numerical simulations are inherently limited by the ability of computers to resolve the range of scales. Our formulation involves describing the underlying quantity at coarser levels of granularity, a common theme that underlies many important applications in physics. We hope that our work will bridge research across generative modeling, simulation, and dynamical systems in related fields.

## Acknowledgments

We greatly thank Google TPU Research Cloud (TRC) for granting us access to TPUs. Zhengyang Geng is partially supported by funding from the Bosch Center for AI. Zico Kolter gratefully acknowledges Bosch's funding for the lab. Mingyang Deng and Xingjian Bai are partially supported by the MIT-IBM Watson AI Lab funding award. We thank Runqian Wang, Qiao Sun, Zhicheng Jiang, Hanhong Zhao, Yiyang Lu, and Xianbang Wang for their help with the implementation of JAX and TPU.

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

# Appendix

## A   Implementation

Table 4: **Configurations on ImageNet 256×256.** B/4 is our ablation model.

| configs | B/4 | B/2 | M/2 | L/2 | XL/2 | XL/2+ |
|---|---|---|---|---|---|---|
| params (M) | 131 | 131 | 308 | 459 | 676 | 676 |
| FLOPs (G) | 5.6 | 23.1 | 54.0 | 80.9 | 119.0 | 119.0 |
| depth | 12 | 12 | 16 | 24 | 28 | 28 |
| hidden dim | 768 | 768 | 1024 | 1024 | 1152 | 1152 |
| heads | 12 | 12 | 16 | 16 | 16 | 16 |
| patch size | 4×4 | 2×2 | 2×2 | 2×2 | 2×2 | 2×2 |
| epochs | 80 | 240 | 240 | 240 | 240 | 1000 |
| batch size | | | | 256 | | |
| dropout | | | | 0.0 | | |
| optimizer | | | | Adam [24] | | |
| lr schedule | | | | constant | | |
| lr | | | | 0.0001 | | |
| Adam $(\beta_1, \beta_2)$ | | | | (0.9, 0.95) | | |
| weight decay | | | | 0.0 | | |
| ema decay | | | | 0.9999 | | |
| ratio of $r \neq t$ | Tab. 1a | | | 25% | | |
| $(r, t)$ cond | Tab. 1c | | | $(t, t - r)$ | | |
| $(r, t)$ sampler | Tab. 1d | | | lognorm(–0.4, 1.0) | | |
| $p$ for adaptive weight | Tab. 1e | | | 1.0 | | |
| CFG effective scale $\omega'$ | Tab. 1f | 2.0 | 2.0 | 2.5 | 2.5 | 2.0 |
| CFG $\omega$, Eq. (21) | $\omega = \omega'$ | 1.0 | 1.0 | 0.2 | 0.2 | 1.0 |
| CFG $\kappa$, Eq. (21) | | | | $\kappa = 1 - \omega/\omega'$ | | |
| CFG cls-cond drop [18] | | | | 0.1 [18] | | |
| CFG triggered if $t$ is in: | [0.0, 1.0] | [0.0, 1.0] | [0.0, 1.0] | [0.0, 0.8] | [0.0, 0.75] | [0.3, 0.8] |

**ImageNet 256×256.** We use a standard VAE tokenizer to extract the latent representations.[4] The latent size is 32×32×4, which is the input to the model. The backbone architectures follow DiT [34], which are based on ViT [8] with adaLN-Zero [34] for conditioning. To condition on two time variables (*e.g.,* $(r, t)$), we apply positional embedding on each time variable, followed by a 2-layer MLP, and summed. We keep the DiT architecture blocks untouched, while architectural improvements are orthogonal and possible. The configuration specifics are in Tab. 4.

**CIFAR-10.** We experiment with class-unconditional generation on CIFAR-10. Our implementation follows standard Flow Matching practice [29]. The input to the model is 32×32×3 in the pixel space. The network is a U-net [38] developed from [44] (∼55M), which is commonly used by other baselines we compare. We apply positional embedding on the two time variables (here, $(t, t - r)$) and concatenate them for conditioning. We do *not* use any EDM preconditioner [22].

We use Adam with learning rate 0.0006, batch size 1024, $(\beta_1, \beta_2) = (0.9, 0.999)$, dropout 0.2, weight decay 0, and EMA decay of 0.99995. The model is trained for 800K iterations (with 10K warm-up [15]). The $(r, t)$ sampler is lognorm(–2.0, 2.0). The ratio of sampling $r \neq t$ is 75%. The power $p$ for adaptive weighting is 0.75. Our data augmentation setup follows [22], with vertical flipping and rotation disabled.

---

[4] https://huggingface.co/pcuenq/sd-vae-ft-mse-flax

| $\kappa$ | FID, 1-NFE |
|-----|-----|
| 0.0 | 20.15 |
| 0.5 | 19.15 |
| 0.8 | 19.10 |
| 0.9 | **18.63** |
| 0.95 | 19.17 |

Table 5: **Improved CFG for MeanFlow**. $\kappa$ is as defined in Eq. (20), whose goal is to enable both class-conditional $u^{\text{cfg}}(\,\cdot\mid\mathbf{c})$ and class-unconditional $u^{\text{cfg}}(\cdot)$ to appear in the target. In this table, we fix the *effective* guidance scale $\omega'$, given by $\omega' = \omega/(1-\kappa)$, as 2.0. Accordingly, for different $\kappa$ values, we set $\omega$ by $\omega = (1-\kappa)\cdot\omega'$. If $\kappa = 0$, it falls back to the CFG case in Eq. (19) (see also Tab. 1f). Similar to standard CFG's practice of randomly dropping class conditions [18], we observe that mixing class-conditional and class-unconditional $u^{\text{cfg}}$ in our target improves generation quality.

# B  Additional Technical Details

## B.1  Improved CFG for MeanFlow

In Sec. 4.2, we have discussed how to naturally extend our method to supporting CFG. The only change needed is to revise the target by Eq. (19). We notice that only the class-*unconditional* $u^{\text{cfg}}$ is presented in Eq. (19). In the original CFG [18], it is a standard practice to *mix* class-conditional and class-unconditional predictions, approached by random dropping. We observe that a similar idea can be applied to our regression target as well.

Formally, we introduce a mixing scale $\kappa$ and rewrite Eq. (16) as:

$$v^{\text{cfg}}(z_t, t\mid\mathbf{c}) = \omega\,v(z_t, t\mid\mathbf{c}) + \kappa\,u^{\text{cfg}}(z_t, t, t\mid\mathbf{c}) + (1-\omega-\kappa)\,u^{\text{cfg}}(z_t, t, t). \tag{20}$$

Here, the role of $\kappa$ is to mix with $u^{\text{cfg}}(\,\cdot\mid\mathbf{c})$ on the right hand side. We can show that Eq. (20) satisfies the original CFG formulation (Eq. (13)) with the *effective guidance scale* of $\omega' = \frac{\omega}{1-\kappa}$, leveraging the relation $v(z_t, t) = v^{\text{cfg}}(z_t, t)$ and $v^{\text{cfg}}(z_t, t) = u^{\text{cfg}}(z_t, t, t)$ that we have used for deriving Eq. (16). With this, Eq. (19) is rewritten as

$$\tilde{v}_t \triangleq \omega\,\underbrace{(\epsilon - x)}_{\text{sample } v_t} + \underbrace{\kappa\,u_\theta^{\text{cfg}}(z_t, t, t\mid\mathbf{c})}_{\text{cls-cond output}} + \underbrace{(1-\omega-\kappa)\,u_\theta^{\text{cfg}}(z_t, t, t)}_{\text{cls-uncond output}}. \tag{21}$$

The loss function is the same as defined in Eq. (17).

The influence of introducing $\kappa$ is explored in Tab. 5, where we fix the effective guidance scale $\omega'$ as 2.0 and vary $\kappa$. It shows that mixing by $\kappa$ can further improve generation quality. We note that the ablation in Tab. 1f in the main paper did *not* involve this improvement, *i.e.,* $\kappa$ was set as 0.

## B.2  Loss Metrics

The squared L2 loss is given by $\mathcal{L} = \|\Delta\|_2^2$, where $\Delta = u_\theta - u_{\text{tgt}}$ denotes the regression error. Generally, one can adopt the powered L2 loss $\mathcal{L}_\gamma = \|\Delta\|_2^{2\gamma}$, where $\gamma$ is a user-specified hyperparameter. Minimizing this loss is equivalent to minimizing an adaptively weighted squared L2 loss (see [14]): $\frac{d}{d\theta}\mathcal{L}_\gamma = \gamma(\|\Delta\|_2^2)^{(\gamma-1)} \cdot \frac{d\|\Delta\|_2^2}{d\theta}$. This can be viewed as weighting the squared L2 loss ($\|\Delta\|_2^2$) by a loss-adaptive weight $\lambda \propto \|\Delta\|_2^{2(\gamma-1)}$. In practice, we follow [14] and weight by:

$$w = 1/(\|\Delta\|_2^2 + c)^p, \tag{22}$$

where $p = 1 - \gamma$ and $c > 0$ is a small constant to avoid division by zero. If $p = 0.5$, this is similar to the Pseudo-Huber loss in [43]. The adaptively weighted loss is $\text{sg}(w) \cdot \mathcal{L}$, where sg denotes the stop-gradient operator.

## B.3  On the Sufficiency of the MeanFlow Identity

In the main paper, starting from the definitional Eq. (4), we have derived the MeanFlow Identity Eq. (6). This indicates that "Eq. (4) $\Rightarrow$ Eq. (6)", that is, Eq. (6) is a necessary condition for Eq. (4). Next, we show that it is also a sufficient condition, that is, "Eq. (6) $\Rightarrow$ Eq. (4)".

In general, equality of derivatives does *not* imply equality of integrals: they may differ by a constant. In our case, we show that the constant is canceled out. Consider a "displacement field" $S$ written as:

$$S(z_t, r, t) = (t - r)u(z_t, r, t). \tag{23}$$

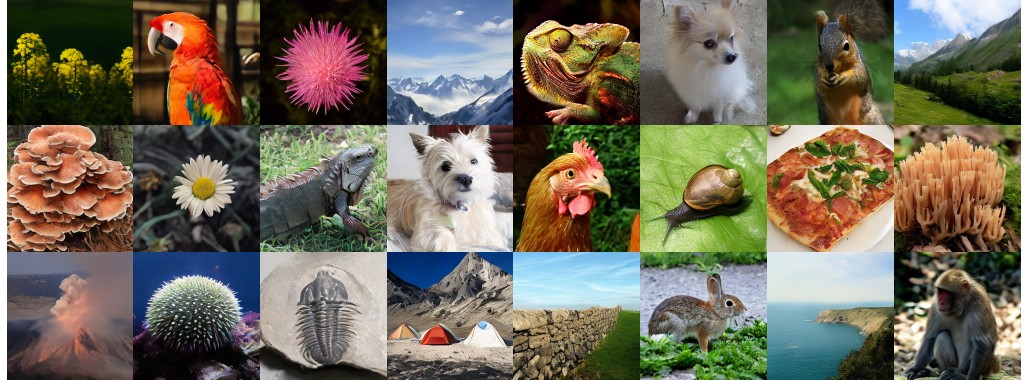

Figure 5: **1-NFE Generation Results**. We show curated examples of class-conditional generation on ImageNet 256×256 using our 1-NFE model (MeanFlow-XL/2, 3.43 FID).

If we treat $S$ as an arbitrary function, then in general, equality of derivatives can only lead to equality of integrals, up to some constants:

$$\frac{d}{dt}S(z_t, r, t) = v(z_t, t) \implies S(z_t, r, t) + C_1 = \int_r^t v(z_\tau, \tau)d\tau + C_2. \tag{24}$$

However, the definition of $S$ gives $S|_{t=r} = 0$, and we also have $\int_r^t vd\tau = 0$ when $t = r$, which gives $C_1 = C_2$. This indicates "Eq. (6) $\Rightarrow$ Eq. (4)".

We note that this sufficiency is a consequence of modeling the average velocity $u$, rather than directly the displacement $S$. Enforcing the equality of derivatives on the displacement, for example, $\frac{d}{dt}S = v$, does *not* automatically yield $S = \int_r^t vd\tau$. If we *were* to parameterize $S$ directly, an extra boundary condition $S|_{t=r} = 0$ is needed. Our formulation can automatically satisfy this condition.

### B.4 Analysis on Jacobian-Vector Product (JVP) Computation

While JVP computation can be seen as a concern in some methods, it is very lightweight in ours. In our case, as the computed product (between the Jacobian matrix and the tangent vector) is subject to stop-gradient (Eq. (10)), it is treated as a constant when conducting SGD using backpropagation *w.r.t.* $\theta$. Consequently, it does not add any overhead to the $\theta$-backpropagation.

As such, the *extra* computation incurred by computing this product is the "backward" pass (formally, forward-mode auto-diff) performed by the jvp operation. This "backward" is analogous to the standard backward pass used for optimizing neural networks (*w.r.t.* $\theta$)—in fact, in some deep learning libraries, such as JAX [3], they use similar interfaces to compute the standard backpropagation (*w.r.t.* $\theta$). In our case, it is even less costly than a standard backpropagation, as it *only* needs to backpropagate to the input variables, not to the parameters $\theta$. This overhead is small.

We benchmark the overhead caused by JVP in our ablation model, B/4. We implement the model in JAX and benchmark on v4-8 TPUs. We compare MeanFlow with its Flow Matching counterpart, where the only *overhead* is the backward pass incurred by JVP; notice that the forward pass of JVP is the standard forward propagation, which Flow Matching (or any typical objective function) also needs to perform. In our benchmarking, the training cost is 0.045 sec/iter using Flow Matching, and 0.052 sec/iter using MeanFlow, which is merely a 16% *wall-clock* overhead.

## C  Qualitative Results

Fig. 5 shows curated generation examples on ImageNet 256×256 using our 1-NFE model.

