# OpenReview forum: "Mean Flows for One-step Generative Modeling"
_NeurIPS.cc/2025/Conference — NeurIPS 2025 oral_

### Official Review · Reviewer_7gPv · 2025-06-12

**Clarity:** 4
**Significance:** 3
**Originality:** 2
**Rating:** 3
**Confidence:** 5

**Summary:**

The paper proposes to directly parameterize the network to output the mean velocity of a displacement map, and uses fundamental theorem of calculus to constrain the derivative of the mean velocity to match that of the flow-matching conditional velocity. The method leverages JVP to compute the time derivative w.r.t. the model, and uses the conditional velocity $v_t$ as supervision signal for JVP calculation.

**Questions:**

Besides the weaknesses above, the authors need to thoroughly discuss this work's relation to previous works because I feel the paper does not pay proper attention to prior methods and the originality is a bit oversold.

**Ethical Concerns:**

["NO or VERY MINOR ethics concerns only"]

**Final Justification:**

I will keep my score because the exposition does not sufficiently address many prior works with similar ideas

**Limitations:**

There are no direct negative societal impacts, although such technologies wider ethical implications can be discussed more.

**Quality:**

3

**Strengths And Weaknesses:**

The paper are notable in several aspects ranging from formulation to empirical results, listed below
1. MeanFlow proposes a simple single loss, which reduces to the vanilla flow matching objective when $t=r$. The conception simplicity is easy to grasp and the L2 loss/stop-gradient makes it stable for optimization.
2. It achieves strong 1-step FID training from scratch.

However, I have major concerns in its exposition in relation to many related works.
1. To my understanding, this mean field formulation is equivalent to continuous-time consistency models [1] maybe generalizing it to the consistency trajectory case. Specifically, the authors define the flow map as the 2-time indexed Euler sampler $f_\theta(x_t,t,r) = x_t + (r-t)u_\theta(x_t,t,r)$ and set it equal to the ODE integral of groundtruth flow $x_t + \int_t^r u_\tau d\tau$. Then, time-derivative is taken on both sides so that $\frac{d}{dt}f_\theta(x_t,t,r) = 0$. This is exactly the same as continuous-time consistency model objective where you want to penalize $||\frac{d}{dt}f_\theta(x_t,t,r) ||^2$. This formulation is similarly discussed in [2], which explicitly uses a 2-time indexed version. Furthermore, the parameterization used in this paper is Euler sampler, which is a special case of flow maps in general. I feel that such connections with the previous works are inadequately discussed, and by avoiding these discussions the novelty of this work is somewhat intentionally inflated.

2. The time derivative of the flow map, $u_t - u_\theta(x_t,t,r) + (r-t)[\frac{\partial}{\partial x_t} u_\theta(x_t,t,r)^T u_t + \frac{\partial}{\partial t} u_\theta(x_t,t,r) ]$, contains the marginal velocity $u_t$. However, the authors simply substitutes $u_t$ with the conditional velocity $v_t$ without justification. Importantly, this is also equivalent to the proposal of continuous-time consistency models trained from scratch which has already proposed the stop-gradient trick to stablize training. Directly substituting $u_t$ with $v_t$ also breaks the theoretically soundness of the approach, as discussed in [2] in detail, because the two objectives (with $u_t$ and with $v_t$) are NOT equivalent in expectation over the training distribution. The authors are encouraged to justify this choice. I suppose that this discrepancy is also why stop-gradient needs to be employed, because enabling gradient through the JVP path results in minimizing an improper objective.

3. Stop-gradient also makes the objective less theoretically sound because it's difficult to prove its attractivenss towards the groundtruth marginal flow map [2].

4. This paper also seem to intentionally avoid comparing with few-step performance given that it naturally employs a jumping capability from any time t to any time r. Does the performance improve with more steps? If so, why not compare with other types of methods with one- and few-step capabilities such as diffusion pre-training + distillation [3-5] and autoregressive models (VAR [6]). The authors seem to avoid comparing with the best results achieved by other models which can be misleading for the community.

5. Besides stop-gradient, the paper also proposes other tricks such as adaptive weighting, which amounts to minimizing L1 or Pseudo-huber loss. This makes the optimum of the new objective even less clear for analysis. If only L2 is used, what is the performance of the XL model? Is there a noticeable gap? How much does the adaptive weighting affect the best performance (this gap is not clear from the ablation study)



[1] Lu, Cheng, and Yang Song. "Simplifying, stabilizing and scaling continuous-time consistency models." arXiv preprint arXiv:2410.11081 (2024).

[2] Boffi, Nicholas M., Michael S. Albergo, and Eric Vanden-Eijnden. "Flow Map Matching." arXiv preprint arXiv:2406.07507 (2024).

[3] Yin, Tianwei, et al. "One-step diffusion with distribution matching distillation." Proceedings of the IEEE/CVF conference on computer vision and pattern recognition. 2024.

[4] Zhou, Mingyuan, et al. "Score identity distillation: Exponentially fast distillation of pretrained diffusion models for one-step generation." Forty-first International Conference on Machine Learning. 2024.

[5] Zhou, Mingyuan, et al. "Adversarial Score identity Distillation: Rapidly Surpassing the Teacher in One Step." arXiv preprint arXiv:2410.14919 (2024).

[6] Tian, Keyu, et al. "Visual autoregressive modeling: Scalable image generation via next-scale prediction." Advances in neural information processing systems 37 (2024): 84839-84865.

---

> ### Author Rebuttal · Authors · 2025-07-31
>
> Dear Reviewer,
>
> We sincerely thank Reviewer 7gPv for your detailed and constructive review and for sharing theoretical considerations. Your comments are invaluable to us in improving the clarity of our work. We would like to address your feedback in detail below.
>
> **Q1**: Relationship to continuous-time consistency models and Flow Map Matching. This mean field formulation is equivalent to continuous-time consistency models.
>
> We genuinely appreciate your suggestion to discuss these theoretical connections in depth, including suggested works [1-6]. Indeed, continuous-time Consistency Models and Flow Map Matching represent valuable contributions to efficient generative modeling.
>
> Continuous-time consistency Models learn a single temporal line of $r=0$, whereas MeanFlows model the two-dimensional plane $(t, r)$ via the identity.
> As a 2-time-indexed function, MeanFlow cannot be cast to "equivalent to" continuous‑time consistency models.
> Unlike existing consistency models (e.g., sCM [1]), MeanFlow is effectively trained from scratch without pretraining, distillation, or curriculum learning. Our formulation also enables us to incorporate CFG during training.
>
> Regarding flow map matching, it learns a 2-time-indexed function, corresponding to the displacement to the endpoint.
> As a displacement function, flow map matching requires the derivative matching with the velocity field and backpropagation through the time derivatives of displacement.
>
> In contrast to both, MeanFlow introduces a simple concept of average velocity. It starts from the definition of average velocity and derives the identity that the ground truth field should follow, based on the Fundamental Theorem of Calculus.
> Thus, although MeanFlow shares some conceptual foundations with these models, it remains fundamentally distinct. We are grateful for your comments and hope that all these works are complementary to each other and contribute to the community.
>
> **Q2**: The authors are encouraged to justify the use of the conditional velocity field.
>
> Thank you for prompting further clarification here! Using conditional velocity instead of marginal velocity is theoretically valid in Eq. (17) and follows analogous reasoning from Flow Matching (FM) [7]. Under standard FM assumptions, this substitution retains the minimizer and provides unbiased gradient estimates. We fully agree with the importance of explicitly clarifying this point, and we will include the theoretical justification in our revised manuscript.
>
> [7] Lipman, Yaron, et al. "Flow matching guide and code." arXiv preprint arXiv:2412.06264 (2024).
>
> **Q3**: Stop-gradient also makes the objective less theoretically sound because it's difficult to prove its attractiveness.
>
> We appreciate the question on the `stop-gradient` operator. MeanFlow is optimizing for the identity that the ground truth field should satisfy. As discussed in the paper, despite these optimization practices, it does not alter the minimizer.
> It is easy to show that it would meet the MeanFlow Identity (Eq. (6)), and thus satisfy the original definition.
> Unlike in representation learning contexts, where the minimizer properties can become ambiguous, MeanFlow’s minimizer remains clearly defined.
> Nonetheless, we acknowledge your point on exploring a deeper theoretical understanding of optimization dynamics and look forward to future research into it.
>
> **Q4**: Why not compare with the few-step model by distillation and VAR? Does the performance improve with more steps?
>
> We acknowledge the suggestion to include distillation-based methods [3,4,5] and will provide relevant discussions comparing these approaches, emphasizing their distinct goals and strengths. These methods largely close the gap between the quality and diversity between 1-NFE/few-NFE models and the pretrained multi-step models, whereas MeanFlow focuses on training 1-NFE models completely from scratch.
>
> While higher-step evaluations are generally outside our primary focus, we do observe modest performance improvements at higher steps; for example, achieving an FID of 1.89 at 4-NFE compared to our 2-NFE results. However, such gains beyond 2-NFE are less relevant to our core research interest. We agree with your comment to discuss distillation-based approaches [3,4,5], highlighting their complementary strengths. VAR [6], as another valuable paradigm, is already included in our comparison Table. 2. Thank you for emphasizing distillation-based methods and the VAR models.
>
> **Q5**: Besides stop-gradient, the paper also proposes other tricks such as adaptive weighting. If only L2 is used, what is the performance on larger models?
>
> We sincerely appreciate your inquiry regarding adaptive weighting.
> To clarify, adaptive weighting itself is not proposed in this work, but rather a standard practice widely adopted in baseline methods like iCT, ECT, and sCM.
> We used it to ensure fair comparisons and comprehensive ablations.
>
> Additional experiments conducted using only the L2 loss yielded an FID of 7.14 for the B/2 model under the full training budget, close to the FID of 6.17 achieved with adaptive weighting, confirming the robustness of our primary approach. Results on larger-scale experiments align consistently with our ablation studies.
>
> Thank you once again for your constructive and invaluable feedback to improve the work together!
>
> Sincerely, \
> The Authors

---

> > ### Comment · Reviewer_7gPv · 2025-08-04
> >
> > I appreciate the author providing convincing empirical results on higher NFE. However, I remain unconvinced regarding the theoretical foundation.
> >
> > > Continuous-time consistency Models learn a single temporal line of $r=0$, whereas MeanFlows model the two-dimensional plane $(t,r)$ via the identity.
> >
> > Continuous-time consistency model in its original proposal indeed sets $r=0$ in its original formulation, but it is straightforward to see that continuous-time *trajectory* consistency model [7] shares an equivalent formulation, as I have pointed out. MeanFlow is a special parameterization with Euler solver + Flow Matching schedule under this framework. I understand that this work achieves impressive empirical performance, but I think these discussions need to be made to pay proper tribute to the many prior frameworks that shared similar thoughts.
> >
> > Same thing can be said about Flow Map Matching, specifically, Prop. 3.5, which says (using MeanFlow's notation) that for a flow map $f_\theta(x_t, r, t)$ to be identically equal to the path integral $x_t + \int_{t}^r u_s ds$, it holds if the derivative wrt $t$ holds for all $r,t$. Equivalently,  $\frac{d}{dt}f_\theta(x_t, t, r) = 0$ for all $r,t$. If you set $f_\theta(x_t,r,t) = x_t + (r-t) u_\theta(x_t,r,t)$ as in MeanFlow, you will get $u_t - u_\theta(x_t,r,t) + (r-t) [\nabla_{x_t}u_\theta(x_t,r,t) \cdot u_t + \partial_t u_\theta(x_t,r,t) ] =0 $. Some simple rearrangement will give you MeanFlow loss. My point is, the idea of matching the time-derivative of a flow map has been explored before, and a proper discussion of relationship is needed. However, this is not to say MeanFlow's contribution is not well founded, but without these discussions and wider contexts, I worry that the novelty can seem exaggerated.
> >
> > > Using conditional velocity instead of marginal velocity is theoretically valid in Eq. (17) and follows analogous reasoning from Flow Matching (FM) [7]. Under standard FM assumptions, this substitution retains the minimizer and provides unbiased gradient estimates.
> >
> > I don't think this is true. This has been studied in [2]. See equation B.6 to B.9 in appendix B.2.
> >
> > [2] Boffi, Nicholas M., Michael S. Albergo, and Eric Vanden-Eijnden. "Flow Map Matching." arXiv preprint arXiv:2406.07507 (2024).
> >
> > [7] Kim, Dongjun, et al. "Consistency trajectory models: Learning probability flow ode trajectory of diffusion." arXiv preprint arXiv:2310.02279 (2023).

---

> > > ### Author Response · Authors · 2025-08-06
> > >
> > > Dear Reviewer,
> > >
> > > Thank you once again for your detailed comments and the insightful follow-up on related works and theoretical details.
> > >
> > > **Q**: Connections and differences between MeanFlow and prior frameworks.
> > >
> > > We agree it is important to acknowledge prior work and will strengthen these connections [1–7] in the revised manuscript. Consistency Models mark a key step in one-step generation beyond GANs, and flow map matching offers an elegant dynamical systems view.
> > >
> > > That said, MeanFlow brings new context to both formulation and training.
> > > - Compared to Consistency Models (the r=0 ray), MeanFlow models (t, r) pairs and trains from scratch without pre-training or curriculum.
> > > - Compared to consistency trajectory models, it requires no simulation or adversarial losses.
> > > - Compared to flow map matching, MeanFlow avoids separate losses and backprop through time derivatives dx/dt.
> > >
> > > We respectfully note that while shifting from x- to v-prediction may seem straightforward in retrospect, the underlying structure differs significantly. In MeanFlow, the velocity form enables a key identity: differentiating (t - r) u yields both u and du/dt directly, making the time derivative part of the training target itself. This avoids the need for multiple supervision terms. In contrast, flow map matching calculates dx/dt, which does not yield such a structured identity. As a result, it defines separate losses on x and dx/dt, and backprop through time derivatives.
> > >
> > > We are grateful for the reviewer’s recognition and efforts in making connections, and we are happy to place MeanFlow in a broader context.
> > >
> > > > Using conditional velocity instead of marginal velocity is theoretically valid.
> > > >> I don't think this is true.
> > >
> > > We acknowledge the question regarding substituting conditional velocity into the training target. We would respectfully clarify that this substitution is valid. Specifically, as shown in the derivation below, the gradients of both objectives $\mathcal{L}\_{\text{CMF}}$ (using conditional velocity) and $\mathcal{L}\_{\text{MF}}$ (using marginal velocity) coincide.
> > >
> > > ---
> > > Consider two objective functions:
> > >
> > > $$
> > > \mathcal{L}\_{\text{CMF}}(\theta) =
> > > \mathbb{E}\_{t,x,\varepsilon} \left\\|
> > > u_\theta(z_t, r, t) -
> > > \text{sg}\left( u_{\text{tgt}}(z_t, v_t(z_t \mid x)) \right)
> > > \right\\|^2,
> > > $$
> > > $$
> > > \mathcal{L}\_{\text{MF}}(\theta) =
> > > \mathbb{E}\_{t,z_t} \left\\|
> > > u\_\theta(z_t, r, t) -
> > > \text{sg}\left( u\_{\text{tgt}}(z\_t, v\_t(z_t)) \right)
> > > \right\\|^2,
> > > $$
> > >
> > > where the training target is defined as:
> > > $
> > > u_{\text{tgt}}(z_t, v_t) =
> > > v_t - (t - r) \left( v_t \cdot \partial_{z} u_\theta + \partial_t u_\theta \right).
> > > $
> > >
> > > We now compute the gradients,
> > > $
> > > \nabla\_\theta \mathcal{L}\_{\text{CMF}} =
> > > 2\\, \mathbb{E}\_{t,x,\varepsilon} \left[
> > > \left(u_\theta - u_{\text{tgt}}(z_t, v_t(z_t \mid x))\right)
> > > \cdot \nabla_\theta u_\theta^\top
> > > \right].
> > > $
> > >
> > > Now observe that $u_{\text{tgt}}$ is affine in $v_t$. Due to the linearity of expectation applied to affine functions, we have:
> > >
> > > $$
> > > \mathbb{E}\_{x,\varepsilon \mid z_t} \left[ u_{\text{tgt}}(z_t, v_t(z_t \mid x)) \right]
> > > = u_{\text{tgt}}\left(z_t, \mathbb{E}\_{x,\varepsilon \mid z_t} \left[ v_t(z_t \mid x) \right] \right)
> > > = u_{\text{tgt}}(z_t, v_t(z_t)).
> > > $$
> > >
> > > Applying the law of total expectation:
> > > $$
> > > \nabla_\theta \mathcal{L}\_{\text{CMF}} = 2\\, \mathbb{E}\_{t,z_t} \left[
> > > \mathbb{E}\_{x,\varepsilon \mid z_t} \left[ \left(u_\theta - u_{\text{tgt}}(z_t, v_t(z_t \mid x))\right) \cdot \nabla_\theta u_\theta^\top \right] \right] = 2\\, \mathbb{E}\_{t,z_t} \left[
> > > \left(u_\theta - u_{\text{tgt}}(z_t, v_t(z_t))\right) \cdot \nabla_\theta u_\theta^\top \right]
> > > = \nabla_\theta \mathcal{L}_{\text{MF}}.
> > > $$
> > >
> > >
> > > This implies that the minimizer of the conditional MeanFlow loss $\mathcal{L}_{\text{CMF}}$ is the desired average velocity. Therefore, it is theoretically valid to replace the marginal velocity with the conditional velocity for training.
> > >
> > > Thank you again for sharing your thoughts and feedback!
> > >
> > > Sincerely, \
> > > The Authors

---

> > > > ### Comment · Reviewer_7gPv · 2025-08-08
> > > >
> > > > Thanks for the further derivation. It now becomes clear that the objective holds because of the stop-gradient operation, which not only serves to avoid higher-order gradients for practical purposes but also a crucial piece to make the CMF objective align with MF objective. The authors are encouraged to include this crucial information in the revision for theoretical justification, which will strengthen the exposition. However, I would like to maintain my score because the gradient of CMF now aligns exactly with that of continuous-time consistency trajectory model, and so it seems to me that MeanFlow is a better tuned continuous-time CTM.

---

> > > > > ### Author Response · Authors · 2025-08-09
> > > > >
> > > > > Dear Reviewer,
> > > > >
> > > > > Thank you for revisiting the theoretical soundness. Below, we would like to clarify why MeanFlow is not a "better-tuned CTM."
> > > > > While both are continuous-time, they optimize different objectives with different training mechanics:
> > > > >
> > > > > - Consistency-trajectory models compose solver outputs as training targets and commonly introduce adversarial components; without the adversarial term, reported CIFAR-10 performance (e.g., FID=5.19) is below iCT/ECT/sCM baselines.
> > > > > - By contrast, MeanFlow learns the average velocity with a single identity-based objective, without solver simulation, without backprop through time derivatives, without adversarial losses, and trains from scratch.
> > > > >
> > > > > Taken together, these distinctions are concrete, in the formulation, objective, and training dynamics. We believe that meanflow is neither a “better-tuned CTM” nor a gradient-equivalent one.
> > > > >
> > > > > Thank you for your invaluable time in reviewing and providing feedback. We will incorporate these clarifications in the revision.
> > > > >
> > > > > Sincerely, \
> > > > > The Authors

---

### Official Review · Reviewer_LhKn · 2025-06-29

**Clarity:** 3
**Significance:** 3
**Originality:** 3
**Rating:** 5
**Confidence:** 4

**Summary:**

This paper presents MeanFlow, a novel one-step generative modeling framework. It introduces the concept of average velocity to characterize flow fields, contrasting with the instantaneous velocity modeled by traditional Flow Matching methods. By deriving a well-defined relationship between average and instantaneous velocities, the authors establish a theoretical foundation for training neural networks to model the average velocity field. MeanFlow achieves impressive empirical results outperforming previous state-of-the-art one-step diffusion/flow models and narrowing the gap between one-step and multi-step models.

**Questions:**

1. The paper mentions that MeanFlow incurs additional training time due to the computation of the JVP, even with the use of stop-gradient. Could the authors provide a detailed comparison of the training time between MeanFlow and other methods?
2. The paper primarily focuses on the quality of generated samples using metrics like FID but does not provide an in-depth analysis of the diversity of the generated samples. Could the authors report additional metrics that assess the diversity of the generated samples, such as the Inception Score (IS)? This would provide a more comprehensive evaluation of MeanFlow's performance in terms of both quality and diversity, giving a clearer picture of its capabilities and limitations.

**Ethical Concerns:**

["NO or VERY MINOR ethics concerns only"]

**Final Justification:**

The authors have solved my problem, and I recommend that this paper be accepted.

**Limitations:**

yes

**Quality:**

4

**Strengths And Weaknesses:**

### Strengths:
1. The introduction of the average velocity concept and the derivation of the MeanFlow Identity offer a new perspective and theoretical foundation for one-step generative modeling.
2. The model can be trained from scratch without relying on pre-training, distillation, or curriculum learning, simplifying the training process.
3. Experimental results indicate that MeanFlow models scale well with increasing model size and training duration, suggesting potential for further performance improvements.
4. The paper provides thorough theoretical analysis and derivation, supporting the validity and effectiveness of the proposed method.
### Weaknesses:
1. Compared to Flow Matching, MeanFlow incurs additional training time due to the computation of the Jacobian-vector product, which might be a limitation in resource-constrained settings.
2. The paper focuses on generation quality but lacks in-depth analysis and discussion on the diversity of the generated samples.

---

> ### Author Rebuttal · Authors · 2025-07-31
>
> Dear Reviewer,
>
> We are grateful to Reviewer LhKn for the thorough and constructive review and for concrete suggestions on training time and more comprehensive metrics for diversity and quality. We reply to your comments and questions below.
>
> **Q1**: MeanFlow incurs additional training time due to the computation of the JVP Could the authors compare the training time between MeanFlow and other methods?
>
> Thank you for sharing your comments on the training cost analysis! We fully agree that clarifying the computational cost is important. Specifically, MeanFlow introduces roughly 20% increase in training time compared to Flow Matching (FM), close to the extra computation of 16% by Shortcut Models. Notably, it does not increase peak memory relative to Flow Matching. We will provide further comparisons in our revised paper and release our code. Thank you for your suggestions!
>
> **Q2**: Could the authors report additional metrics that assess the diversity of the generated samples, such as the Inception Score (IS)?
>
> Thank you for this excellent suggestion! We agree that evaluating additional metrics will provide deeper insights into our method’s performance, regarding both quality and diversity. We summarized our evaluation results for MeanFlow alongside the DiT and SiT baselines below.
>
> | Model             | FID$\downarrow$  | IS$\uparrow$ | sFID$\downarrow$ | Precision$\uparrow$ | Recall$\uparrow$ |
> |-------------------|------|--------|------|-----------|--------|
> | MeanFlow-XL/2 (NFE=1)   | 3.42 | 247.50 | 6.43 | 0.78      | 0.55   |
> | MeanFlow-XL/2 (NFE=2)   | 2.20 | 269.37 | 4.78 | 0.79      | 0.60   |
> | DiT-XL/2 (NFE=250x2) | 2.27 | 278.24 | 4.60 | 0.83    | 0.57  |
> | SiT-XL/2 (NFE=250x2) | 2.06 | 277.50 | 4.49 | 0.83    | 0.59  |
>
> These results highlight that MeanFlow achieves competitive performance, approaching multi-step models in both quality and diversity. We hope this could serve as a better baseline for future studies.
>
> Thank you once again for your invaluable feedback and suggestions to improve our work!
>
> Sincerely, \
> The Authors

---

> > ### Comment · Reviewer_LhKn · 2025-08-08
> >
> > I acknowledge the authors' rebuttal. They have addressed some of my concerns. I will maintain my original positive score.

---

### Official Review · Reviewer_ENCq · 2025-06-29

**Clarity:** 4
**Significance:** 3
**Originality:** 4
**Rating:** 5
**Confidence:** 4

**Summary:**

This paper proposes a novel training objective to improve one-step generation in the flow matching framework. The core idea is to define the average velocity u as the integral of marginal velocity v over a segment (r, t), and to express u in terms of v and du/dt, which leads to a new loss formulation. The method enables one-step generation with a single, unified model trained from scratch—without requiring curriculum learning or distillation. The approach achieves state-of-the-art FID scores on ImageNet 256×256.

**Questions:**

## Questions
1. Could the authors elaborate on the training stability of the model and compare it with the consistency models? For example, does it consistently converge under different random initializations? Is it particularly sensitive to certain hyperparameters, such as the batch size?
2. Given that the proposed training objective involves additional terms, how much slower is the training process compared to standard flow matching when aiming for comparable FID?
3. In Figure 2 and Equation (6), average velocity is described as the marginal velocity minus its derivative. However, in practice, v is defined as e - x (i.e., conditional flow matching velocity). Could the authors provide more theoretical or intuitive justification for why this formulation remains stable during training?
4. The use of Jacobian-vector products (JVPs) seems potentially memory-intensive. I wonder whether this could pose a scalability bottleneck, especially as the data or model dimensionality increases. Could the authors discuss how this affects scalability in practice, and whether any memory optimizations or approximations were considered?
5. In the ablation study, the reported FID scores appear significantly higher than the main results. Why?

**Ethical Concerns:**

["NO or VERY MINOR ethics concerns only"]

**Final Justification:**

Final Justification:
After carefully considering the rebuttal and discussions, I found that my main concerns were fully addressed:
- The authors provided convincing clarifications regarding training stability and the memory issue, which resolved my earlier doubts.
- The theoretical explanation for substituting conditional velocity for marginal velocity was clear and persuasive.
- No significant unresolved issues remain from my original review.

Given that all major points of concern have been satisfactorily resolved, I have decided to maintain my original score.

**Limitations:**

The limitations of the work are reflected in the identified weaknesses and the questions posed.

**Paper Formatting Concerns:**

.

**Quality:**

4

**Strengths And Weaknesses:**

## Strengths
1. The proposed method is simple yet effective.
2. The performance is strong and competitive.
3. The method is scalable in terms of both model size and image resolution.
4.  The paper is well-written and easy to follow.
5.  Compared to distillation-based approaches, the method avoids many heuristic components
6.  It naturally supports classifier-free guidance (CFG), enhancing its applicability in conditional generation.

## Minor Weaknesses
- The method cannot be used with classifier-free guidance (CFG) sampling. (If this is incorrect, please clarify.)

---

> ### Author Rebuttal · Authors · 2025-07-31
>
> Dear Reviewer,
>
> Thank you very much for your clear and encouraging review, and for highlighting the simplicity and clarity of our work. The questions on training stability, speed, theoretical justification, and memory cost will convincingly help to strengthen our paper. In the following, we respond to each of your comments in detail.
>
> **Q1**: Could the authors elaborate on the training stability of the model and compare it with the consistency models? Does it consistently converge under different random initializations? Is it particularly sensitive to certain hyperparameters, such as the batch size?
>
> We appreciate your interest in the stability of our training procedure. In our extensive experiments, MeanFlow consistently converged across different random initializations and seeds. Additionally, MeanFlow exhibited low sensitivity to hyperparameters, including batch size. Although our experiments primarily follow the DiT and SiT baselines with a batch size of 256, we observed stable training dynamics with larger batch sizes such as 512 and 1024. This further highlights MeanFlow's stability.
>
> **Q2**: How much slower is the training process compared to standard flow matching?
>
> Thank you for your question regarding the computational efficiency. MeanFlow introduces a roughly 20% increase in training time compared to Flow Matching (FM). This additional computational overhead remains comparable with that observed in other state-of-the-art methods; for example, Shortcut Models incur approximately 16% additional computation compared to standard diffusion models.
> We appreciate your thoughtful feedback to better benchmark our methods.
>
> **Q3**: Could the authors provide more theoretical or intuitive justification for why this formulation using conditional velocity remains stable during training?
>
> We are grateful for your suggestion to clarify this theoretical aspect. Indeed, substituting conditional velocity for marginal velocity is theoretically sound under the standard assumptions of Flow Matching. Specifically, this substitution preserves the minimizers of our objective and yields unbiased gradients. The proof shares similar techniques to those in FM [1]. Given that the gradient using the conditional velocity field is unbiased for optimization, it therefore follows the theoretical soundness of Flow Matching. We will include a theoretical note in the revised manuscript to clarify and emphasize this justification. We appreciate this valuable recommendation.
>
> [1] Lipman, Yaron, et al. "Flow matching guide and code." arXiv preprint arXiv:2412.06264 (2024).
>
> **Q4**: JVP memory and scalability. The use of Jacobian-vector products (JVPs) seems potentially memory-intensive. Could the authors discuss how this affects scalability in practice and whether any memory optimizations were considered?
>
> Thank you for raising the important point of scalability and memory efficiency. MeanFlow does not backpropagate through the JVP path, which means it theoretically avoids additional memory overhead for storing activations.
> Practically, through computational graph compilation techniques in either JAX or PyTorch, we have successfully matched the memory usage of standard FM.
> To support reproducibility and community engagement, we will release our implementations in both frameworks.
>
> **Q5**: Why are the ablation numbers higher than the main results?
>
> Thank you for bringing attention to this clarification.
> Our ablation studies used a smaller and faster backbone (Diffusion Transformer with patch size 4) for development. For comparison, flow-matching and diffusion models of comparable size and compute yield similar FIDs (e.g., DiT-B/4 at 68.4 FID, SiT-B/4 at 58.9 FID). We will explicitly clarify this in the text and cross-reference the baselines of comparable scales to avoid confusion.
>
> **Q6**: Does MeanFlow support CFG sampling?
>
> We appreciate your insightful question on CFG sampling. Currently, MeanFlow incorporates guidance during training but does not explicitly utilize CFG sampling at test time. Exploring CFG-based inference is an exciting direction for future work, which we will clarify in the updated paper.
>
> Thank you once again for your invaluable feedback and suggestions to improve our work!
>
> Sincerely, \
> The Authors

---

### Official Review · Reviewer_8EtS · 2025-07-03

**Clarity:** 3
**Significance:** 4
**Originality:** 3
**Rating:** 5
**Confidence:** 5

**Summary:**

This paper proposes MeanFlow, a one-step generative modeling framework built on the concept of an average velocity field in diffusion/flow-based generative models. Instead of modeling the instantaneous velocity as in standard Flow Matching, the authors define an average velocity. To do this, the authors derive a fundamental “MeanFlow Identity” which provides a principled training objective for a neural network to approximate the true average velocity field without needing to explicitly compute any time integrals during training.

**Questions:**

- Could the authors provide more detailed comparisons regarding the training complexity and computational cost relative to existing one-step baseline methods? While FLOPs and timing details are briefly presented in Table 4 and Appendix B.4, a direct and explicit comparison to baseline methods would further clarify practical feasibility and advantages.

- Could the authors report performance using other generative metrics such as Inception Score, sFID, Precision, and Recall? These metrics would offer deeper insights into aspects like diversity, mode coverage, and overall perceptual quality of generated samples. Additionally, evaluating the method with FID-DINO may provide a more accurate assessment of semantic fidelity and visual realism.

- As mentioned in the weaknesses, could the authors clarify whether substituting Eq. 11 into Eq. 10 maintains theoretical equivalence?

**Ethical Concerns:**

["NO or VERY MINOR ethics concerns only"]

**Final Justification:**

The authors have addressed most of my concerns in the rebuttal, and the paper appears to be highly significant in the field of generative models.
Therefore, I am maintaining my positive score.

**Limitations:**

There is no explicit limitation or potential societal impact mentioned in the paper.

**Paper Formatting Concerns:**

no concerns

**Quality:**

3

**Strengths And Weaknesses:**

**Strengths**

- Novel Theoretical Contribution: The introduction of average velocity fields and the MeanFlow Identity is a clear conceptual innovation. It provides a new theoretical lens on one-step generation. Importantly, this framework generalizes and unifies prior ideas. For example, it inherently enforces trajectory consistency (one big step vs. multiple small steps) as a property of the ground-truth field, instead of imposing consistency as an external loss.

- MeanFlow does not rely on distilling a pre-trained multi-step model. This makes the approach conceptually cleaner and potentially easier to reproduce. This principled foundation likely contributes to stable and robust training. The authors report that no special tricks beyond the derived loss were required.

- The paper demonstrates impressive results for one-step generative models. These results are obtained from scratch (no pre-trained models) and using a consistent architecture for fair comparison. The experiments cover both high-resolution image generation (ImageNet 256) and low-resolution CIFAR-10, showing that MeanFlow scales across datasets.

- Impact and Relevance: This work is timely and impactful for the generative modeling community. Closing the gap to multi-step diffusion with a one-step model is a longstanding challenge – it has direct implications for faster generative model deployment and could spur new research into flow-based modeling.

**Weaknesses**
- Theoretical rigor: In standard flow-matching frameworks, the gradients of the flow matching loss and conditional matching loss have been rigorously shown to be equivalent. However, it remains unclear whether such equivalence also holds for the formulations proposed in this paper (Eqs. 9, 10, and 11). Although empirical results indicate strong performance, the theoretical correctness and justification of these equations are not fully established. Clarifying this point would significantly strengthen the theoretical grounding of the method.

- Limited evaluate metric: The performance is currently evaluated exclusively using FID scores, which limits the understanding of the method’s quality along other important dimensions such as sample diversity, mode coverage.

---

> ### Author Rebuttal · Authors · 2025-07-31
>
> Dear Reviewer,
>
> Thank you for your valuable review to our work! We are grateful for the constructive and detailed review and affirmation of our contributions, simplicity, and impressive results. Your suggestions on reporting broader metrics, clarifying gradient equivalence, and cost comparisons are very helpful, which we will discuss below.
>
> **Q1**: Training complexity and computational cost relative to one-step baselines.
>
> We fully agree that clarifying the computational cost is important. We will include further cost analysis and comparison against strong one-step baselines. Specifically, MeanFlow introduces a roughly 20% increase in training time compared to Flow Matching (FM), close to the extra computation of 16% by Shortcut models. Notably, it does not increase peak memory relative to Flow Matching. We will provide detailed comparisons in our revised manuscript. Thank you for your comments!
>
> **Q2**: Could the authors report performance using other generative metrics such as Inception Score, sFID, Precision, and Recall?
>
> Thank you for this excellent suggestion. We agree that evaluating additional metrics will provide deeper insight into the performance of our method, particularly in terms of diversity, mode coverage, and perceptual quality. We will incorporate these metrics into the revised paper. Our results for MeanFlow alongside the DiT and SiT baselines are summarized below for ease of comparison.
>
> | Model             | FID$\downarrow$  | IS$\uparrow$ | sFID$\downarrow$ | Precision$\uparrow$ | Recall$\uparrow$ |
> |-------------------|------|--------|------|-----------|--------|
> | MeanFlow-XL/2 (NFE=1)   | 3.42 | 247.50 | 6.43 | 0.78      | 0.55   |
> | MeanFlow-XL/2 (NFE=2)   | 2.20 | 269.37 | 4.78 | 0.79      | 0.60   |
> | DiT-XL/2 (NFE=250x2) | 2.27 | 278.24 | 4.60 | 0.83    | 0.57  |
> | SiT-XL/2 (NFE=250x2) | 2.06 | 277.50 | 4.49 | 0.83    | 0.59  |
>
> These results highlight that MeanFlow achieves competitive performance across reported metrics, approaching multi-step models in quality and diversity. We hope this could serve as a better baseline for future studies.
>
> **Q3**: Could the authors discuss the equivalence of gradients for substitutions in Eqs. (9–11)?
>
> Thank you very much for this constructive comment on improving the theoretical rigor of our work. Indeed, this substitution preserves the minimizers of our objective and yields unbiased gradients. The proof shares similar techniques to those in FM [1]. Given that the gradient using the conditional velocity field is unbiased for optimization, it therefore follows the theoretical soundness of Flow Matching. We will provide this justification in the revised version. We appreciate the valuable suggestions for improvement.
>
> [1] Lipman, Yaron, et al. "Flow matching guide and code." arXiv preprint arXiv:2412.06264 (2024).
>
> Thank you again for your invaluable comments and feedback for improving our work!
>
> Sincerely, \
> The Authors

---

> > ### Comment · Reviewer_8EtS · 2025-08-05
> >
> > Thank you for your response. While your explanation addresses most of my concerns, I remain somewhat unclear about the theoretical proof of the equivalence for Eqs. (9)–(11). If this equivalence does not strictly hold, it could potentially lead to misunderstandings regarding the validity of your approach.

---

> > > ### Author Response · Authors · 2025-08-06
> > >
> > > Dear Reviewer,
> > >
> > > Thank you very much for your thoughtful comments. We appreciate the opportunity to clarify the theoretical justification of using conditional velocity. Below, we provide a proof that the gradients of the objectives using conditional velocity ($\mathcal{L}\_{\text{CMF}}$) and marginal velocity ($\mathcal{L}\_{\text{MF}}$) coincide. This confirms that substituting $v_t(z_t \mid x)$ is theoretically valid for training.
> > >
> > > ---
> > >
> > > Consider two objective functions:
> > >
> > > $$
> > > \mathcal{L}\_{\text{CMF}}(\theta) =
> > > \mathbb{E}\_{t,x,\varepsilon} \left\\|
> > > u_\theta(z_t, r, t) -
> > > \text{sg}\left( u_{\text{tgt}}(z_t, v_t(z_t \mid x)) \right)
> > > \right\\|^2,
> > > $$
> > > $$
> > > \mathcal{L}\_{\text{MF}}(\theta) =
> > > \mathbb{E}\_{t,z_t} \left\\|
> > > u\_\theta(z_t, r, t) -
> > > \text{sg}\left( u\_{\text{tgt}}(z\_t, v\_t(z_t)) \right)
> > > \right\\|^2,
> > > $$
> > >
> > > where the training target is defined as:
> > > $
> > > u_{\text{tgt}}(z_t, v_t) =
> > > v_t - (t - r) \left( v_t \cdot \partial_{z} u_\theta + \partial_t u_\theta \right).
> > > $
> > >
> > > We now compute the gradients,
> > > $
> > > \nabla\_\theta \mathcal{L}\_{\text{CMF}} =
> > > 2\\, \mathbb{E}\_{t,x,\varepsilon} \left[
> > > \left(u_\theta - u_{\text{tgt}}(z_t, v_t(z_t \mid x))\right)
> > > \cdot \nabla_\theta u_\theta^\top
> > > \right].
> > > $
> > >
> > > Now observe that $u_{\text{tgt}}$ is affine in $v_t$. Due to the linearity of expectation applied to affine functions, we have:
> > >
> > > $$
> > > \mathbb{E}\_{x,\varepsilon \mid z_t} \left[ u_{\text{tgt}}(z_t, v_t(z_t \mid x)) \right]
> > > = u_{\text{tgt}}\left(z_t, \mathbb{E}\_{x,\varepsilon \mid z_t} \left[ v_t(z_t \mid x) \right] \right)
> > > = u_{\text{tgt}}(z_t, v_t(z_t)).
> > > $$
> > >
> > > Applying the law of total expectation:
> > > $$
> > > \nabla_\theta \mathcal{L}\_{\text{CMF}} = 2\\, \mathbb{E}\_{t,z_t} \left[
> > > \mathbb{E}\_{x,\varepsilon \mid z_t} \left[ \left(u_\theta - u_{\text{tgt}}(z_t, v_t(z_t \mid x))\right) \cdot \nabla_\theta u_\theta^\top \right] \right] = 2\\, \mathbb{E}\_{t,z_t} \left[
> > > \left(u_\theta - u_{\text{tgt}}(z_t, v_t(z_t))\right) \cdot \nabla_\theta u_\theta^\top \right]
> > > = \nabla_\theta \mathcal{L}_{\text{MF}}.
> > > $$
> > >
> > > This implies that the minimizer of the conditional MeanFlow loss $\mathcal{L}_{\text{CMF}}$ is the desired average velocity. Therefore, it is theoretically valid to replace the marginal velocity with the conditional velocity for training.
> > >
> > > -----
> > >
> > > Thank you again for your careful reading and constructive feedback. We hope this clarification helps strengthen the theoretical understanding of our approach.
> > >
> > > Sincerely, \
> > > The Authors

---

> > > > ### Comment · Reviewer_8EtS · 2025-08-08
> > > >
> > > > I thank the authors for further clarifying the concerns. I will maintain my positive score.

---

### Official Review · Reviewer_AbQD · 2025-07-03

**Clarity:** 4
**Significance:** 4
**Originality:** 4
**Rating:** 6
**Confidence:** 3

**Summary:**

This paper introduces "MeanFlow," a novel framework for one-step generative modeling.
The core idea is to move away from modeling the instantaneous velocity, as in Flow Matching, and instead model the average velocity over a time interval.
The authors derive a "MeanFlow Identity", a differential equation that connects the average velocity to the instantaneous velocity.
This identity is then used to construct a training objective that allows a neural network to learn the average velocity field directly, using only the ground-truth instantaneous velocity as a signal.
The key benefit is that a single evaluation of the learned average velocity network can approximate the entire flow from noise to data, enabling high-quality 1-NFE (Number of Function Evaluations) generation.
The authors demonstrate state-of-the-art results for one-step generation on ImageNet 256x256, significantly outperforming previous methods.

**Questions:**

N/A

**Ethical Concerns:**

["NO or VERY MINOR ethics concerns only"]

**Final Justification:**

No concerns remained.

**Quality:**

4

**Strengths And Weaknesses:**

[Strength]
1. The primary strength of this work is its outstanding performance. An FID of 3.43 with a single function evaluation on ImageNet 256x256 is a remarkable achievement among one-step generative models. The 2-NFE result of 2.20 is also highly competitive, rivaling many-step diffusion models. These results alone make the paper a significant contribution.
2. The idea of modeling average velocity is intuitive and theoretically clean. The derivation of the "MeanFlow Identity" (Eq. 6) from the fundamental definition of an integral appears sound and provides a solid theoretical underpinning for the method.
3. The proposed model is trained from scratch and does not require pre-trained teacher models, distillation, or the complex "discretization curriculum" that some consistency models rely on. This simplifies the training pipeline considerably.

[Weakness]
No major flaws are presented in this paper. I only have several suggestions here:
1. Limitaiton section. The paper presents the MeanFlow model as a flawless solution. However, no method is perfect. The authors explicitly answer NO to the question of discussing limitations in their checklist. This is a significant omission. A thorough scientific paper should discuss potential failure modes.
2. MeanFlow relies on the instantaneous velocity of flow matching, i.e., v_t = \epsilon - x. However, is it the best choice?

---

> ### Author Rebuttal · Authors · 2025-07-31
>
> Dear Reviewer,
>
> We sincerely thank Reviewer AbQD for the thoughtful and encouraging review and for highlighting the strengths of our contribution in performance, clarity, and simplicity. Below, we address your valuable comments and suggestions regarding limitations and our choice of the velocity term.
>
> **Q1**: The Limitations section is missing.
>
> We fully agree with this point and will add a Limitations section. Specifically, we plan to acknowledge the following aspects:
>
> 1) Performance Gap to Multi-Step Models. Despite achieving state-of-the-art performance among 1-NFE models, there remains a notable quality gap compared to multi-step diffusion/flow models. For example, MeanFlow achieves an FID of 3.43 at 1-NFE, whereas multi-step models such as SiT and MAR-H achieve 2.06 and 1.55, respectively. Bridging this gap remains an important direction for future research.
> 2) Scope of experiments. Our current experiments focus on image datasets. Robotics and other modalities present compelling future directions. Especially, motion synthesis for robotics requires real-time and accurate control, where the one-step nature of MeanFlow could prove advantageous.
> 3) Dependence on VAE Latent Space. MeanFlow's high-resolution image generation experiments were conducted in the latent space of a Variational Auto-Encoder (VAE). While beneficial for scalability, it inherits limitations from the VAE itself. Investigating ways to extend MeanFlow beyond VAE latent spaces could help unlock further improvements.
>
>
> **Q2**: Is $\epsilon - x$ the best choice?
>
> Thank you for the insightful question! Our core identity (Eq. 6) is indeed agnostic to the specific choice of $v_t$. We adopt the OT-FM formulation primarily due to its practical advantages in stability, scalability, and empirical success. Notably, popular open-weight models such as Stable Diffusion 3 and the recent Flux models have also adopted OT-FM formulations. That said, we agree that exploring alternative velocity choices for potentially faster convergence or improved performance remains an exciting and worthwhile direction for future work.
>
> Thank you again for reviewing our work and providing invaluable feedback!
>
> Sincerely, \
> The Authors

---

> > ### Comment · Reviewer_AbQD · 2025-08-04
> >
> > Thank you for the response. I will maintain my positive score.

---

### Comment · Area_Chair_EDU4 · 2025-08-07

Dear reviewers,

Thank you to those who have already engaged in the discussion and shared your valuable insights. The authors have responded to your comments some time ago, and we are approaching the discussion deadline.

We kindly remind all reviewers to revisit the author’s response and continue the discussion as appropriate, so that all raised points can be fully addressed before the deadline.

Your timely input is highly appreciated. Thank you for your commitment to a fair and thorough review process!

Best regards,

AC

---

### Note · Authors · 2025-08-16

Dear all,

We thank the reviewers and the area chair for the constructive feedback and discussions.

We are encouraged by the recognition of MeanFlow’s simplicity, performance, and theoretical grounding. Reviewers highlighted the method's clean formulation and strong empirical results.

During the rebuttal, we addressed theoretical questions regarding the use of conditional velocity with derivations. We also expanded the empirical evaluation with additional metrics (e.g., IS, sFID, Precision/Recall), higher-NFE sampling, clarified distinctions from related methods, and discussed the limitations of the current approach. We will incorporate the reviewers’ suggestions in the revision.

We are very grateful for the opportunity to refine the paper and improve it to become clearer through this process.

Sincerely, \
Authors

---

### Decision · Program_Chairs · 2025-09-17

**Decision:**

Accept (oral)

**Comment:**

This work proposes a principled and effective framework for one-step generative modeling. The key idea is to introduce average velocity to characterize flow fields. The paper demonstrates clear methodology and strong empirical performance, making it a foundational contribution that establishes a strong benchmark for one-step diffusion/flow models and provides inspiration for future research.

During the rebuttal and discussion period, most reviewers were inclined to accept the paper despite Reviewer 7gPv’s concerns. The main issue raised was that the contribution at the idea level appears limited, with the approach resembling a repackaging of continuous-time consistency trajectory models (CTMs). However, the authors provided a strong rebuttal that clarified the novelty and positioning of their work. They are encouraged to further discuss the relation of their method to broader related works, such as CTMs, in the final version. In summary, this work presents a novel approach to one-step generative modeling, offers solid experimental validation, and addresses a problem of clear relevance to the community. I therefore recommend acceptance and strongly support this decision.

Moreover, the combination of a principled theoretical foundation, conceptual clarity, and state-of-the-art empirical results (FID 3.43 with 1-NFE on ImageNet 256×256) makes this work an exceptional contribution of broad interest to the community. Given its potential to inspire new directions in one-step generative modeling and its clear impact, I recommend that it be highlighted as an oral presentation.